# A Deep Learning Approach for Diagnosis Support in Breast Cancer Microwave Tomography

**DOI:** 10.3390/diagnostics13101693

**Published:** 2023-05-10

**Authors:** Stefano Franceschini, Maria Maddalena Autorino, Michele Ambrosanio, Vito Pascazio, Fabio Baselice

**Affiliations:** 1Department of Engineering, University of Napoli Parthenope, Centro Direzionale, 80143 Napoli, Italy; mariamaddalena.autorino001@studenti.uniparthenope.it (M.M.A.); vito.pascazio@uniparthenope.it (V.P.); fabio.baselice@uniparthenope.it (F.B.); 2Department of Economics, Law, Cybersecurity, and Sports Sciences, University of Napoli Parthenope, Via della Repubblica 32, 80035 Nola, Italy; michele.ambrosanio@uniparthenope.it

**Keywords:** microwave tomography, breast cancer detection, neural networks, biomedical imaging, artificial intelligence, electromagnetic inverse scattering

## Abstract

In this paper, a deep learning technique for tumor detection in a microwave tomography framework is proposed. Providing an easy and effective imaging technique for breast cancer detection is one of the main focuses for biomedical researchers. Recently, microwave tomography gained a great attention due to its ability to reconstruct the electric properties maps of the inner breast tissues, exploiting nonionizing radiations. A major drawback of tomographic approaches is related to the inversion algorithms, since the problem at hand is nonlinear and ill-posed. In recent decades, numerous studies focused on image reconstruction techniques, in same cases exploiting deep learning. In this study, deep learning is exploited to provide information about the presence of tumors based on tomographic measures. The proposed approach has been tested with a simulated database showing interesting performances, in particular for scenarios where the tumor mass is particularly small. In these cases, conventional reconstruction techniques fail in identifying the presence of suspicious tissues, while our approach correctly identifies these profiles as potentially pathological. Therefore, the proposed method can be exploited for early diagnosis purposes, where the mass to be detected can be particularly small.

## 1. Introduction

Microwave tomography is an interesting imaging modality, able to provide qualitative and quantitative information of biological tissues in a noninvasive and safe way. In its typical configuration, an array of antennas surrounds the anatomical district under investigation. Due to the interaction between electromagnetic field and biological tissues, electromagnetic waves are scattered and, consequently, acquired by the antennas. These waves are nonionizing and, therefore, safe for the subjects’ health. Considering a Multiple-Input-Multiple-Output (MIMO) tomographic system, it is possible to collect signals related to different points of view and to exploit these data to reconstruct the electric properties of the considered tissues (i.e., permittivity and conductivity).

Cancer is the second leading cause of death in the United States, following heart diseases. The coronavirus disease 2019 (COVID-19) pandemic caused delays in the diagnosis and treatment of cancer because of health care setting closures, disruptions in employment and health insurance, and fear of COVID-19 exposure. Among all sites, breast cancer represents the most common type of cancer in women (in the United States, almost 300,000 estimated cases in 2023) [1]. The mortality of this type of cancer is lower than 15% due to modern screening techniques enabling early diagnosis and treatments. Therefore, the availability of imaging techniques capable of identifying very small neoplastic tissues is of paramount importance for medical prevention. At present, most of the breast cancer diagnosis techniques are based on X-ray Mammography, Magnetic Resonance Imaging (MRI), UltraSound scanning (US), and γ-ray Nuclear Medicine [2].

Relatively to X- and γ-rays, these methodologies can achieve impressive results in terms of spatial resolution (e.g., mammography can reach a pixel size of the order of 50/100 microns [3]). Nevertheless, both methodologies involve the use of ionizing radiations, thus limiting the exposure time for patients due to health issues. Furthermore, mammography requires breast compression, resulting in uncomfortable exams, with images affected by low contrast and grainy problems. Additionally, nuclear imaging presents slow imaging times and very high costs [4].

In MRI nonionizing radiations are exploited, and therefore, this technique is more recommended since it is not harmful for patient’s health. Unfortunately, despite a very good spatial resolution, this exam is expensive and cumbersome. Moreover, since a magnetic field is involved, this technique is unfeasible for patients with metallic implants.

US is a noninvasive and safe imaging modality. The resulting images are characterized by a good resolution but the medical examination has some limitations, mainly related to the presence of speckle noise and its operator-dependent nature.

Microwave tomography for breast cancer imaging is expected to solve some of the aforementioned issues, and, thus, the research community has focused on this technology in recent years. The microwave system should be effective in breast cancer detection as physiological and pathological tissues present different electrical properties, both in terms of permittivity and conductivity [5]. It is important to underline that the information acquired by these systems is complementary with respect to those acquired by other systems, such as the tissue attenuation coefficient in the case of X-ray mammography or the acoustical impedance variation in the case of US imaging.

In its typical configuration, microwave tomographic scanners adopt a set of antennas (transmitters and receivers) surrounding the breast, generally with a matching medium filling the imaging region [6,7,8,9,10,11,12,13,14,15,16,17,18,19]. Recently, some microwave tomographic imaging systems started their clinical trials, showing their potentialities in supporting medical diagnosis [20,21,22].

After the acquisition step, algorithms for image formation are exploited. Their role consists of solving an electromagnetic inverse scattering (EIS) problem and producing the permittivity and the conductivity maps of the imaged tissues. In its general formulation, this problem is ill-posed and nonlinear due to multiple interactions in the tissue propagation [23,24,25]. Therefore, most efforts of the research community in recent years has focused on the development of effective imaging algorithms. One of the key aspects is the identification of proper approximations to reduce the problem complexity, e.g., Born and Kirchhoff approximations [26]. More sophisticated approaches, generally implementing iterative procedures, have also been proposed. Such techniques operate without any approximation, but at the cost of high computational complexity, being unsuitable for real-time applications.

In this framework, algorithms based on Artificial Intelligence (AI) are attracting the attention of several research groups. Artificial Neural Networks (ANNs) and, in more detail, Deep Learning (DL), have already demonstrated their potentialities in classification, segmentation, and regression tasks in several fields, such as computer vision [27], short-range radars [28,29], and remote sensing [30]. Concerning clinical applications, Neural Networks (NNs) have been exploited in numerous imaging problems involving medical devices such as MRI, CT, and US [31]. Inspired by these excellent results, various research groups have developed DL solutions in the EIS framework [32].

Initially, AI techniques have been considering for retrieving essential information regarding targets shape and size [33]. More recently, DL was applied for retrieving permittivity and conductivity maps of the investigated scene. Most of the approaches do not provide an NN for an end-to-end solution, e.g., reconstruction of the electric properties of breast tissues starting from the scattering matrix. For instance, in [34,35,36], NNs are exploited either as a regularization strategy or for obtaining super-resolution. Other techniques consider NNs for improving recovery performance and obtaining more accurate reconstructions starting from raw dielectrics maps provided by direct inversion methods, such as back-propagation [37]. In this framework, Convolutional Neural Networks (CNNs) are typically adopted. In more detail, CNNs architectures composed of an encoder and a decoder path were exploited for quantitative reconstructions [38]. This CNN-based topology is known as *U-net* and was firstly introduced for segmentation tasks. Recent advancements have been provided by [39], where a Generative Adversarial Network (GAN) has been applied to raw versions of the dielectric profiles to obtain a more accurate solution. In a previous study, our group provided a fully-connected ANN-based approach for end-to-end quantitative breast electric properties reconstruction [40,41,42]. A realistic breast phantom database was considered for testing the algorithm, showing promising results. Conversely, from the other studies where deep-learning-based imaging techniques have been described, the neural network proposed in this manuscript has a different scope. The idea consists of exploiting a neural network for detecting the presence of a neoplasm by processing the collected scattered field data without performing the imaging. By moving from an estimation to a detection problem, the proposed methodology is expected to achieve the capability of detecting even tiny neoplastic masses, whose size is smaller than conventional resolution limits, and enabling early breast cancer diagnosis.

The manuscript is organized as follows. Section 2 illustrates the proposed approach, while the performance assessment of the methodology and some numerical experiments are reported in Section 3. Finally, some conclusions end the manuscript.

## 2. Methodology

A sketch of the considered acquisition scheme is shown in Figure 1, in which a ring of antennas surrounds the breast. The system works by transmitting a Radio Frequency (RF) signal from one antenna at a time, and receiving the scattered waves with all the others. At the end of the scan, the collected data form the so-called *scattering matrix*. The acquisition model can be expressed by the following equations [43]:(1)$$\begin{array}{cc} E_{s} & =A_{e}\left(\epsilon^{\prime },\sigma ,E_{t}\right), \end{array}$$(2)$$\begin{array}{cc} E_{t} & =E_{i}+A_{i}\left(\epsilon^{\prime },\sigma ,E_{t}\right), \end{array}$$

In Equations (1) and (2), $E_{i}$, $E_{s}$, and $E_{t}$ represent the incident, the scattered, and the total electric fields, respectively, $A_{i}$ and $A_{e}$ are the radiating operators, and $\epsilon^{\prime }$ and σ refer to the relative permittivity and conductivity of the scene, respectively.

The aim of quantitative imaging algorithms is the estimation of $\epsilon^{\prime }$ and σ profiles by processing the acquired scattering matrix. The clinical outcome is related to the identification of neoplastic regions characterized by higher permittivity and conductivity values. In this paper, a neural-network-based procedure was designed to discriminate between healthy and suspicious breasts without forming the images of $\epsilon^{\prime }$ and σ. The idea is to identify the presence of regions with high (over the physiological range) values of dielectric permittivity and conductivity directly from the scattering matrix. A sketch of the proposed methodology is reported in Figure 2. The core of the proposed algorithm is a neural network made of 10 convolutional layers, each with 32 features and a 3 × 3 filter dimension. The network receives as input the scattering matrix and estimates as output the probability that a pathological tissue is present. Thresholding this value leads to the classification of the imaged profile as physiological or pathological.

In order to evaluate quantitatively the performance of the proposed classification algorithm, an ad hoc training and testing database was considered. In this paper, a 2D scenario is assumed. Similarly to [40], a random shape generator was exploited for the definition of the breast inner geometry [44] and, subsequently, three healthy breast tissues were simulated: *fibro-glandular*, *transitional* and *adipose*. In order to analyze pathological profiles, tumors of random size were added within the fibro-glandular regions. Each profile can contain one or more masses with size in the range [0.2, 9] cm^2^. The breast outer shape is approximated by an ellipse with axes in the range [6.5, 12] cm with an orientation in the range [0, 2π].

As a further step, the permittivity and conductivity maps were generated according to the statistical distributions estimated from the database in [45] and reported in Table 1. Regarding tumors, dielectric characteristics are assumed to range uniformly in [57, 63] in case of relative permittivity and [1.1, 1.3] S/m for the conductivity. Based on fibro-glandular percentage, it is possible to define four breast classes. More in detail, breasts with a fibro-glandular percentage in the range [5–20]% belong to class A. Class B refers to breasts with a fibro-glandular percentage in the range [20, 30]%. Percentages in a range of [30, 40]% belong to class C. Finally, in class D, breasts fibro-glandular percentage is in the range [40, 65]% are included. Every breast profile is placed in a background lossless medium with permittivity equal to 15. The imaging domain is 15 × 15 cm^2^, discretized in a grid of 108 × 108 pixels (resulting in a pixel size is 0.14 × 0.14 cm^2^). An example of both real and imaginary parts of the complex permittivity of a profile belonging to class B is shown in Figure 3.

For each profile, the scattering matrix was computed via a Fast Fourier Transform-Conjugate Gradient (FFT-CG) forward solver based on the Method of Moments (MoM) [24], assuming a multiview-multistatic system with transmitters and receivers located in 30 angular equally spaced locations on a measurement circle of radius 12 cm. The transmitted RF signal is a 1 GHz sine wave. The scattering matrix includes complex numbers, and therefore, the input of the algorithm is a 3D matrix of size 30 × 30 × 2, considering both the amplitude and phase of each scattered field measure. The output of the neural network is a score in the range [0, 1] representing the probability for the considered profile of being potentially pathological.

A database composed of 160,000 profiles (20,000 per each class in both healthy and pathological cases) was considered for both training and testing the network. The database was partitioned to use 80% of the profiles for training, 10% for validation, and the remaining 10% for testing.

To train the network, a cross-entropy cost function was exploited. An adaptive moment estimation method (ADAM) was employed with an initial learning rate of $5\times 10^{-5}$ and 100 epochs for the training. Regarding the computational time, the training phase took about 30 min on a 64 bit workstation with an AMD Ryzen 3990X CPU and an NVIDIA Quadro RTX 6000 GPU, while the processing of a single scattering matrix is almost instantaneous.

## 3. Results

In order to evaluate quantitatively the performance of the proposed approach, different metrics and indicators were adopted. The proposed approach basically provides a score related to the probability for the considered slice to be suspicious. Therefore, a crucial parameter for the approach is the threshold $T_{h}$, which is paramount for the classification of each profile as healthy or not. Thus, for each working point (i.e., $T_{h}$ value), different rates can be computed. Defining $T_{p}$ as the number of correct suspicious assignments, $T_{n}$ as the number of correct healthy assignments, $F_{p}$ as the number of wrong suspicious assignments, and $F_{n}$ as the number of wrong healthy assignments, it is possible to calculate the following quantities:(3)$$\begin{array}{cc} \mathrm{True}\;\mathrm{Positive}\;\mathrm{Rate}\;\left(\mathrm{TPR}\right) & =\frac{T_{p}}{T_{p}+F_{n}}, \end{array}$$(4)$$\begin{array}{cc} \mathrm{False}\;\mathrm{Positive}\;\mathrm{Rate}\;\left(\mathrm{FPR}\right) & =\frac{F_{p}}{F_{p}+T_{n}}, \end{array}$$

As well as the complementary quantities: (5)$$\begin{array}{cc} \mathrm{False}\;\mathrm{Negative}\;\mathrm{Rate}\;\left(\mathrm{FNR}\right) & =1-TPR=\frac{F_{n}}{T_{p}+F_{n}}, \end{array}$$(6)$$\begin{array}{cc} \mathrm{True}\;\mathrm{Negative}\;\mathrm{Rate}\;\left(\mathrm{TNR}\right) & =1-FPR=\frac{T_{n}}{F_{p}+T_{n}}, \end{array}$$

The quantities TPR and TNR are also known as *Sensitivity* and *Specificity*, respectively.

Based on the TPR and FPR values, it is possible to compute the Receiver Operative Characteristic (ROC) curve [46]. In Figure 4, ROC curves related to the test set described in Section 2 for different values of SNR are shown. From the figure, it is possible to note that the performance of the system decrease considerably when the SNR is lower than 20 dB.

From each ROC curve, several synthetic metrics were extracted, i.e., the Area Under ROC (AUR), representing the overall performance of the receiver—the closer this value is to 1 the better the system performance—and the Equal Error Rate (EER), defined as the threshold which gives an FPR equal to FNR—in this case, values closer to 0 represent better performance. In addition, fixing the value of $T_{h}$ (in our test fixed at 0.5), it is possible to compute the accuracy, sensitivity, and specificity. Table 2 reports the values of AUR, EER, accuracy, sensitivity, and specificity.

Similarly to what is shown in Figure 4, Table 2 confirms that for SNR values greater than 30 dB, the results are excellent. Additionally, considering sensitivity and specificity, it is possible to note that, for low values of SNR, the approach is unbalanced, since sensitivity values are greater than specificity values.

In the framework of breast cancer diagnosis, the detection of small tumors is paramount for early detection and diagnosis. Therefore, in the following, the capability of the system to detect suspicious profiles containing small tumors is evaluated. In particular, from the overall testing dataset, a sub-dataset composed of profiles belonging to class B with only a single tumor of size in the range [0.2, 2.8] cm^2^ is considered. According to the tumor extension, this sub-dataset is divided in seven sets and the related EER and AUR of the system, as shown in Figure 5 (due to space limitations, only the 30 dB case is reported).

From the figure, it is possible to note that the system is quite stable independently from the size of the tumor. This result is quite encouraging, specifically for early detection purposes. To highlight this concept, Figure 6 and Figure 7 show the reconstructions of dielectric properties of two breast profiles. More in detail, the approaches considered for these reconstructions belong to different families of inverse solvers. The first one is the distorted Born Iterative Method (DBIM) [47], the second is the Contrast Source Inversion (CSI) [48], and, finally, a reconstruction provided by a DL method is considered [40].

In all the numerical simulations, the adopted frequency is 1 GHz and a total of 30 antennas act as transmitters and receivers in a multiview-mustitatic fashion surrounding the imaging domain. To perform the single-frequency nonlinear inversion via the local minimization schemes related to the DBIM and CSI approaches, a convenient initial guess was used, i.e., a homogeneous phantom with the shape of the reference profile (assumed a priori known as morphological information) and with electric features equal to the average properties of the adipose tissue.

Relatively to the DBIM implementation, a conjugate-gradient least-square linear inversion approach was adopted at each DBIM iteration to perform the inversion and to generate the update. To further reduce the ill-posed quality of the EIS problem under consideration, a spatial projection on a coarser grid was adopted in addition to the use of the support information.

On the other hand, concerning the CSI implementation, a Cross-Correlated CSI (CC-CSI) was adopted, as detailed in [49]. In this solution, the performance of the classic CSI approach is improved by considering a modified cost function, which interrelates the mismatch of the state equation and the data error in the measurement space.

Regarding the DL approach of [40], the fully-connected NN described in the paper has been originally trained with only healthy profiles. In order to perform reconstructions of potentially pathological scenarios, a fine-tuning procedure considering the dataset described in Section 2 was employed.

From both figures, it is possible to note that if the size of the tumor is sufficiently big, all the reconstruction approaches can correctly retrieve its morphological and electric properties. Conversely, when the tumor size is particularly small, none of the proposed reconstruction methods can correctly locate the mass. It is worth noting that the classification method proposed in this manuscript managed to classify both the profiles correctly as suspicious with a score *P* equal to 1.

## 4. Conclusions

In this paper, a deep learning approach for diagnosis support in microwave breast cancer imaging was proposed. The method takes as input measures of the electromagnetic fields acquired by antennas placed around the breast and provides as output a score related to the probability of the breast to have malignant tissues.

The algorithm was tested by exploiting an ad hoc realistic database with different levels of noise and with masses of different sizes. The results are quite encouraging, showing good potentialities for the proposed approach, in particular for those cases where the tumor is small. As a matter of the fact, the proposed approach predicts correctly the presence of abnormalities even when conventional approaches fail to identify suspicious masses. These results support the thesis that the proposed method might be a good candidate in early breast cancer diagnosis where the abnormalities are small and it is nontrivial to be detected by conventional methods. It is worth underlining that the processing time is almost instantaneous, as opposed to most of the imaging algorithms, which might be time consuming.

Further research will focus on the design of advanced methods to retrieve additional information regarding suspicious masses (e.g., morphology), while simultaneously preserving the excellent accuracy of the proposed approach.

## Figures and Tables

**Figure 1 diagnostics-13-01693-f001:**
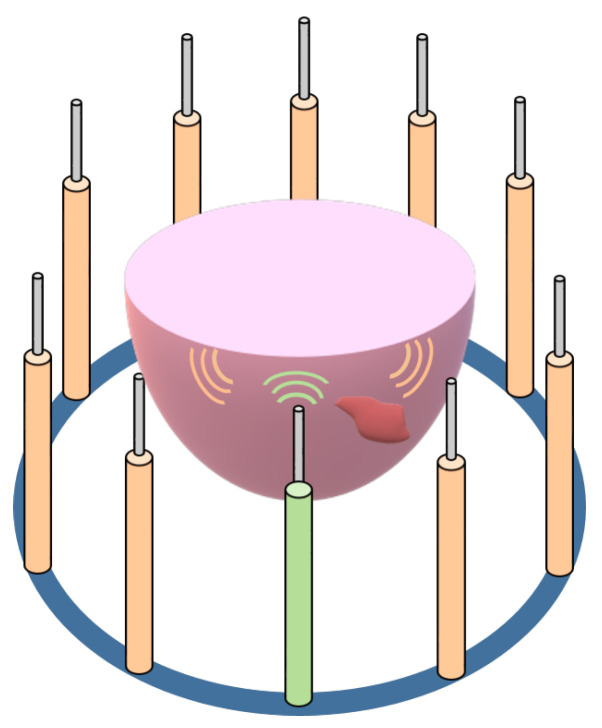
Sketch of a microwave tomography system. The antennas surround objects under investigation. Changing the active antenna, it is possible to obtain information related to different views.

**Figure 2 diagnostics-13-01693-f002:**
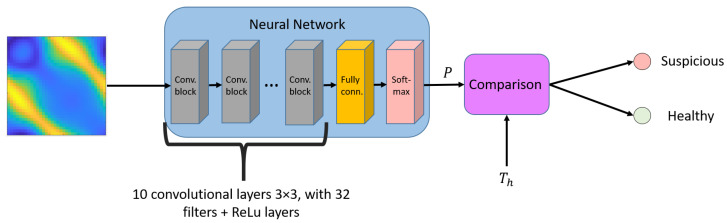
Block diagram of the proposed approach. The scattering matrix is analyzed by a neural network classifier. The binary output is the class of the profile: *suspicious* or *healthy*.

**Figure 3 diagnostics-13-01693-f003:**
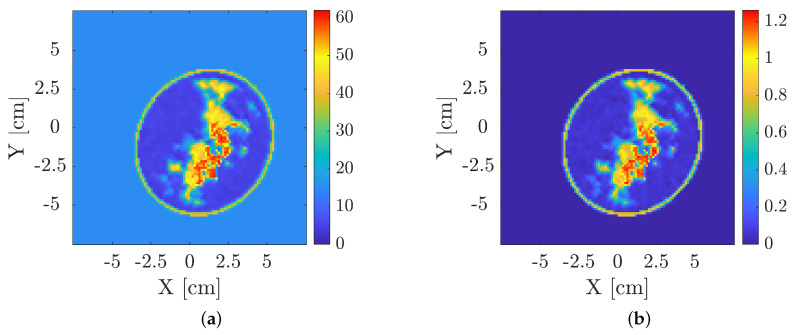
Example of a breast profile exploited for the training and testing of the approach (class B). Relative permittivity is shown in (**a**), conductivity (S/m) in (**b**). Red regions refer to malignant areas.

**Figure 4 diagnostics-13-01693-f004:**
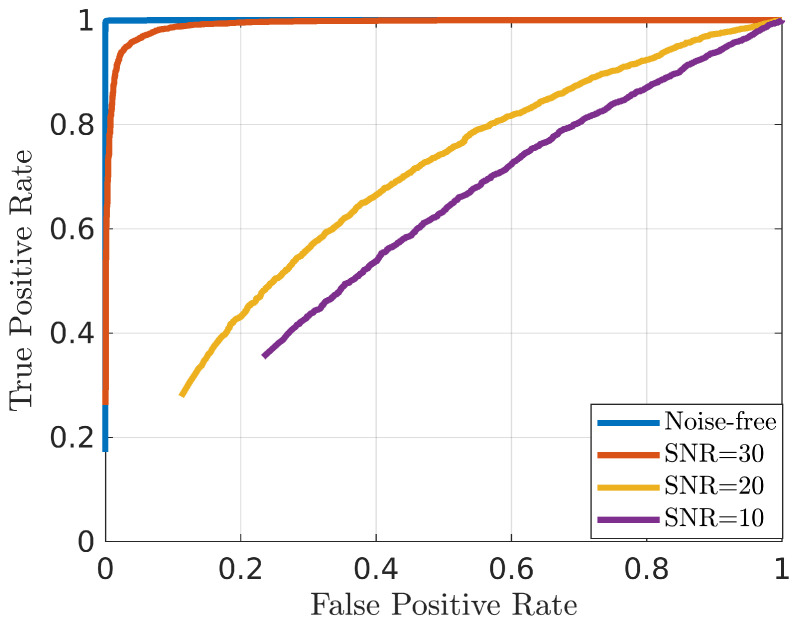
ROC curve of the proposed approach for a testing set with different levels of noise.

**Figure 5 diagnostics-13-01693-f005:**
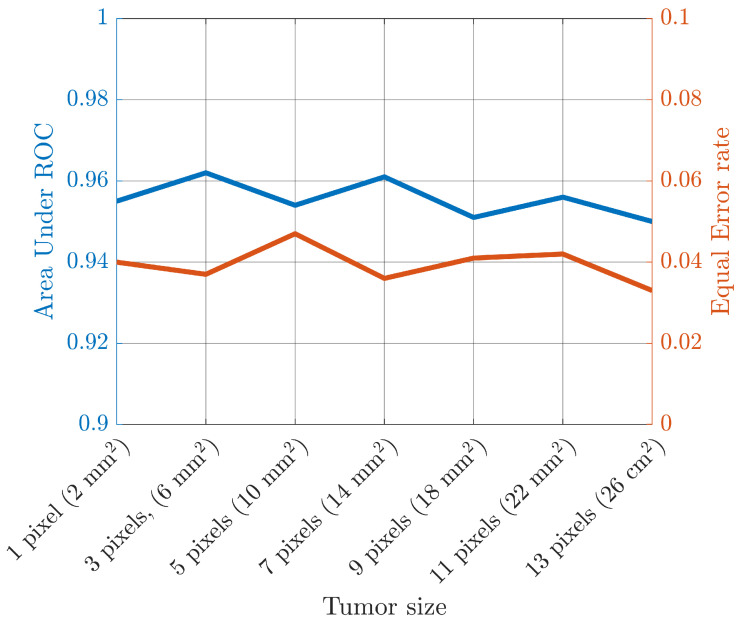
Area under ROC and equal error rate of the proposed approach for a testing set with different tumor sizes.

**Figure 6 diagnostics-13-01693-f006:**
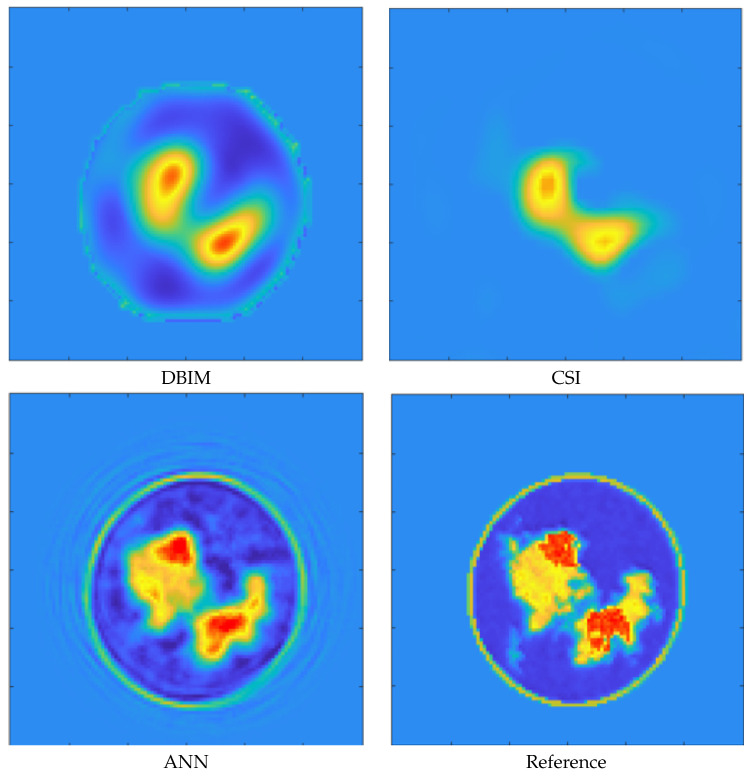
“Big” tumor case. Relative permittivity map reconstructions obtained by a DBIM, CSI, and ANN method. Reference profiles are shown in the bottom right image. The proposed classification method presents a *P* score equal to one for this profile.

**Figure 7 diagnostics-13-01693-f007:**
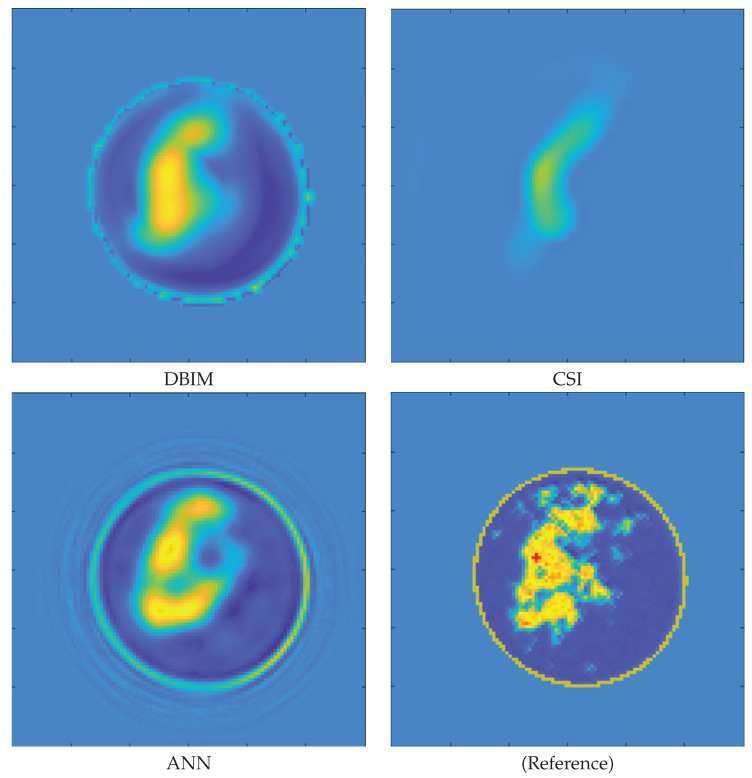
“Small” tumor case. Relative permittivity map reconstructions obtained by a DBIM, CSI, and ANN method. Reference profiles are shown in the bottom right image. The proposed classification method presents a *P* score equal to one for this profile.

**Table 1 diagnostics-13-01693-t001:** Dielectric properties of healthy breast tissues.

Tissue Type	Relative Permittivity	Conductivity [S/m]
Adipose	1–8	0–0.12
Transitional	7–38	0.11–0.58
Fibroglandular	37–58	0.56–1.22

**Table 2 diagnostics-13-01693-t002:** Performance of the proposed approach for the testing set corrupted by different levels of noise.

	Noise-Free	30 dB SNR	20 dB SNR	10 dB SNR
**Area under ROC**	1.00	0.992	0.661	0.550
**Equal Error Rate**	0.001	0.041	0.367	0.430
**Accuracy**	0.995	0.960	0.625	0.567
**Sensitivity**	0.989	0.950	0.738	0.626
**Specificity**	0.999	0.950	0.502	0.495

## Data Availability

All the data are available on request to the authors.

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
