# Peer review of "A Deep Learning Approach for Diagnosis Support in Breast Cancer Microwave Tomography"

_diagnostics, 2023, doi:10.3390/diagnostics13101693_

Round 1

Reviewer 1 Report

This paper shows very good results for imaging generation employing IA aiming at breast cancer detection. 

Would be great more detail about the solution applied. 

The big challenge for my comprehension is the RF realm. The paper mentions a sine wave of 1 GHz. What is the system bandwidth? How does it discuss resolution without bandwidth information? How about RF leakage? What kind of antennas were employed? 

To refer to microwave imaging as relatively new is very subjective. Recommend to re-write it.

Mammography is the highest-resolution modality for medical imaging. Recommend to re-write about mammography.

Some acronyms need to be revised and start with a capital letter.

Reviewer 2 Report

In the article the deep learning technique for tumor detection in microwave tomography is proposed.

Remarks.

1. It is unclear what is the underlying mathematical model?

2. How is the scattering matrix measured? How many frequency sets are used? What is the probing signal? Frequency or time domain?

3. What are the electromagnetic parameters of healthy tissue? What happens at the border of healthy and diseased tissue?

4. The conjugate-gradient least-square linear inversion approach was adopted at each DBIM iteration to perform the inversion and to generate the update. It is better to clarify what mathmodel was solved numericaly, what mesh and so on.

The article can be published after answering the questions.

Author Response

Please see te attached file.

Round 2

Reviewer 1 Report

No more suggestions.

Reviewer 2 Report

Thank you for reply!

To detail the network, it is desirable to provide a detailed diagram (the composition can be downloaded using Python development tools).
